# Anti-Inflammatory, Antidiabetic Properties and In Silico Modeling of Cucurbitane-Type Triterpene Glycosides from Fruits of an Indian Cultivar of *Momordica charantia* L.

**DOI:** 10.3390/molecules26041038

**Published:** 2021-02-16

**Authors:** Wilmer H. Perera, Siddanagouda R. Shivanagoudra, Jose L. Pérez, Da Mi Kim, Yuxiang Sun, Guddadarangavvanahally K. Jayaprakasha, Bhimanagouda S. Patil

**Affiliations:** 1Vegetable and Fruit Improvement Center, Department of Horticultural Sciences, Texas A&M University, 1500 Research Parkway, Suite A120, College Station, TX 77843, USA; Wilmer.Perera@gmail.com (W.H.P.); siddu4191@gmail.com (S.R.S.); ppmd_4@tamu.edu (J.L.P.); gkjp@tamu.edu (G.K.J.); 2Department of Nutrition, Texas A&M University, College Station, TX 77843, USA; dmkim5322@tamu.edu (D.M.K.); yuxiangs@tamu.edu (Y.S.)

**Keywords:** *Momordica charantia*, cucurbitane-type triterpene glycosides, charantoside XV, α-amylase, α-glucosidase, anti-inflammatory activity, in silico study

## Abstract

Diabetes mellitus is a chronic disease and one of the fastest-growing health challenges of the last decades. Studies have shown that chronic low-grade inflammation and activation of the innate immune system are intimately involved in type 2 diabetes pathogenesis. *Momordica charantia* L. fruits are used in traditional medicine to manage diabetes. Herein, we report the purification of a new 23-*O*-β-d-allopyranosyl-5β,19-epoxycucurbitane-6,24-diene triterpene (charantoside XV, **6**) along with 25ξ-isopropenylchole-5(6)-ene-3-*O*-β-d-glucopyranoside (**1**), karaviloside VI (**2**), karaviloside VIII (**3**), momordicoside L (**4**), momordicoside A (**5**) and kuguaglycoside C (**7**) from an Indian cultivar of *Momordica charantia*. At 50 µM compounds, **2**–**6** differentially affected the expression of pro-inflammatory markers *IL-6*, *TNF-α*, and *iNOS*, and mitochondrial marker *COX-2*. Compounds tested for the inhibition of α-amylase and α-glucosidase enzymes at 0.87 mM and 1.33 mM, respectively. Compounds showed similar α-amylase inhibitory activity than acarbose (0.13 mM) of control (68.0–76.6%). Karaviloside VIII (56.5%) was the most active compound in the α-glucosidase assay, followed by karaviloside VI (40.3%), while momordicoside L (23.7%), A (33.5%), and charantoside XV (23.9%) were the least active compounds. To better understand the mode of binding of cucurbitane-triterpenes to these enzymes, in silico docking of the isolated compounds was evaluated with α-amylase and α-glucosidase.

## 1. Introduction

The genus *Momordica* L., with around 59 species, is one of the most abundant genera in the Cucurbitaceae family. *Momordica charantia* L. is a taxon extensively studied for its antidiabetic properties [1]. Two varieties have been taxonomically identified: *M. charantia* L. (var. *muricata* (Willd.) Chakrav. and var. *charantia*) [1]. The *muricata* variety produces small size fruits, whereas *charantia* typically produces larger fruits [1]. Several phenotypes of *M. charantia* var. *charantia* appear in nature, displaying noteworthy differences in fruit size and shape. The most common fruit from the Indian cultivar has a greenish color, a narrower shape, and a serrated surface. *M. charantia* var. *charantia* L. is commonly consumed in traditional Asian cuisines and is well-known for its unique taste and flavor [2,3].

Cucurbitane-type triterpenes are the most representative sub-class of compounds in *M. charantia*, and some investigations suggest that they are responsible for their antidiabetic property [4,5]. More than 270 cucurbitane-type triterpenes have been isolated from different plant organs from 1980 to date. However, these compounds are found in very low concentration and their composition has been reported to vary between cultivars. Our comprehensive survey of the literature indicates that the Indian phenotype has received less attention than the Chinese or Sri Lanka phenotypes. Octonorcucurbitacins A, B and C (342 Da) are the smallest cucurbitane-type compounds reported from *M. charantia,* whereas momordicoside T (1110 Da) the largest one with four sugar units, one glucose linked at position C-25, two glucose units, and one xylose attached at C-3 as follows: -Glc[Glc(1-4)]Xyl(1–4) [4,5,6]. Additionally, several aglycones, mono, di, and tri-glycoside triterpene derivatives have also been isolated from bitter melon. The glycosidic moieties attached to these triterpenes typically consist of D series of allose, galactose, glucose and xylose monosaccharide units, mainly at positions C-23 and C-25 of the side chain, and C-3 and C-7 of the A and B rings, respectively [4,7,8,9].

Many purified compounds, as well as crude extracts, have been screened in response to various ethnomedicinal claims, concluding that bitter melon may play a potential role in the management of various chronic diseases by working as an antidiabetic, antioxidant, antiviral, antiobesity, and anticancer agent among others [8,9,10,11,12,13,14]. Moreover, some studies have been conducted in fruit, seeds, and leaves showing hypoglycemic activity in both diabetic animal and human models [15,16]. Thus, *M. charantia* var. *charantia* stands out as a promising natural alternative to reduce the risk and/or manage diabetes mellitus type II [17].

Diabetes is a chronic disease that affects millions of people worldwide. Therefore, it is vital to find new natural prevention and management strategies for this disease and related complications. Results from our previous studies indicated that cucurbitane-type triterpenoids isolated from Chinese cultivar might modulate biological activities involved in the pathogenicity of diabetes [18,19]. Due to these molecules structural complexity, much more work is warranted to test biological activities and better understand the structure-activity relationship. Therefore, herein we describe the isolation and structure elucidation of a new cucurbitane-type triterpene together with six known compounds from acetone extract of the Indian cultivar of *M. charantia*. The anti-inflammatory activity of purified compounds using lipopolysaccharide (LPS)-activated RAW264.7 macrophage cells, inhibition of the α-amylase, and α-glucosidase enzymes were presented. Furthermore, the molecular interaction of purified compounds with α-amylase and α-glucosidase enzymes was also studied using *in silico* molecular docking.

## 2. Results and Discussion

### 2.1. Isolation of Compounds

The isolation of the compounds presented was based on two main approaches, silica gel and reversed-phase chromatographic separations. Compound **6** was purified as a white amorphous powder, [α]^25^_D_-71 (*c* 0.1, MeOH). HRESIMS of compound **6** showed a molecular ion at *m*/*z* 687.3967 [M + Na]^+^ (calculated 687.4047 ± 0.008 *m*/*z* [M + Na]^+^ for C_37_H_60_O_10_). The molecular formula of compound **6** was compared with reported triterpenes isolated from *M. charantia* and our in-house database [20]. No match was observed, suggesting an unknown compound. The ^13^C-NMR spectrum of compound **6** showed resonances of 37 carbons. The 2D-gHMQC spectrum revealed the presence of sixteen methines, seven methylenes, seven methyl, one methoxy, and six quaternary carbons. Six tertiary groups appeared at *δ*_H_(*δ*_c_) 0.84 (14.8); 1.74 (26.7); 1.77 (19.1); 1.18 (20.9); 0.87 (24.6); 0.87 (20.6) corresponding to CH_3_-18, CH_3_-26, CH_3_-27, CH_3_-28, CH_3_-29 and CH_3_-30, respectively and one secondary *δ*_H_(*δ*_c_) 0.96 (14.7) (d, *J* = 6.4 Hz, CH_3_-21). The methoxy group was observed at *δ*_H_(*δ*c) 3.41 s, 3H, assigned as H-1″ (58.2, C-1″), one acetal group at *δ*_H_(*δ*c) 4.70 s, 1H, assigned as H-19 (113.3, C-19), three olefinic protons at *δ*_H_(*δ*c) 5.98 [dd, *J* = 9.8 Hz, 2.1 Hz, H-6] (132.2, C-6), *δ*_H_(*δ*c) 5.58 [dd, *J* = 9.8 Hz, 3.4 Hz, H-7] (133.8, C-7) and *δ*_H_(*δ*c) 5.24 [d, *J* = 10.2 Hz, H-24] (124.6, C-24), and one quaternary olefin carbon *δ*_C_ 137.4 (C-25). One anomeric proton at *δ*_H_(*δ*c) 4.73 [d (*J* = 8 Hz, 1H] (102.6) with a beta linkage corresponding to one hexose appeared in the spectrum together with their respectively five oxygenated signals *δ*_H_(*δ*c) 3.32, H-2′ (73.1, C-2′), 4.05, H-3′ (73.0, C-3′), 3.48, H-4′ (68.5, C-4′), 3.58, H-5′ (75.7, C-5′) and 3.60, 3.76, H-6′ (63.1, C-6′). ^1^H and ^13^C NMR chemical shifts are shown in Table 1.

Acid hydrolysis of **6** furnished d-allose, which was identified by comparison of the HPLC retention times of thiocarbamoyl thiazolidine derivative with sugar thiocarbamoyl thiazolidine derivative standards [21]. The NMR data of compound **6** suggested the presence of a 5β,19-epoxycucurbitane triterpenes with a methoxy group and a sugar moiety attached [22]. The position of the methoxy group was established through ^3^*J* HMBC correlations between *δ*_H_ 3.41 ppm (OCH_3_) and the acetal carbon at position C-19 (*δ*c 113.3), indicating that the methoxy group was attached at C-19.

The relative position of the monosaccharide attachment was confirmed by the assignment of gHMBC spectrum. ^3^*J* HMBC correlations between anomeric proton H-1′ (4.73 ppm) and C-23 (81.5 ppm) confirmed the attachment of the sugar at C-23. The position of proton H-2′ of the allose was confirmed through the ^1^H-^1^H COSY correlation between H-2′ (4.05 ppm) and the anomeric proton H-1′ (4.73 ppm). TOCSY correlations were helpful to identify protons corresponding to the sugar moiety and HMBC, and ^1^H-^1^H COSY to assign remaining proton and carbon chemical shifts. The HMBC correlations between C-24 and CH_3_-26 and CH_3_-27; C-25 and H-23, C-22 and H-23 together with C-20 and H-22, suggested that the side chain was -CH(CH_3_)CH(OH)CH(O-All)=C(CH_3_)_2_. Crucial HMBC and ^1^H-^1^H COSY correlations are shown in Figure 1A. This type of side-chain has been described previously for karavilosides IV, VIII, IX and X, 23-*O*-β-allopyranosyl-cucurbita-5,24-dien-7α,3β, 22(*R*), 23(*S*)-tetraol 3-O-*β*-allopyranoside; momordicosides M, N and O isolated from *M. charantia* cultivated in Sri Lanka and China [7,23,24,25]. The stereochemistry of the stereocenters C-22 and C-23 was established by NOESY experiment. A correlation of H-22 was observed with H-21, while no correlation was found for H-22 and H-23 (22S and 22R). C-19 configuration was deduced as *R* due to the correlation between proton H-19 with H-1 and H-2, only observed in *R* configuration (Figure 1B).

This result was also supported by a comparison of the NMR chemical shifts of H-8, C-8, C-9, C-10, C-11, and C-19 previously reported for several 5β,19*R-* and *S*-epoxycucurbitane triterpenes with different substituents attached at C-19 (methoxy, ethoxy, and hydroxyl groups) [21,26]. However, the main difference in chemical shifts was observed for H-8 and C-8: *R* configuration of H-8 range from 2.82 to 3.37 ppm and from 41 to 43 ppm for C-8, whereas for *S* configuration H-8 (2.13–2.32 ppm) and C-8 (49.5–50.1 ppm). Chemical shifts for compound **6** appeared at H-8 (2.86 ppm) and C-8 (43.2 ppm), supporting the *R* configuration of C-19. Hence, the structure of compound **6** was established as 19(*R*)-methoxy-5β,19-epoxycucurbita-6,24-diene-3β,22*S*,23*R*-triol-23-*O*-β-d-allopyranoside (charantoside XV) and shown in Figure 2.

Compounds **1**–**5** and **7** were identified as known monoglycosides by spectroscopic (^1^H- and ^13^C-NMR spectra) and spectrometric analyses (HRESIMS), and by comparison with the data reported in the literature as follows: 25ξ-isopropenylchole-5(6)-ene-3-*O*-β-d-glucopyranoside (**1**), karaviloside VI (**2**), karaviloside VIII (**3**), momordicoside L (**4**), momordicoside A (**5**) and kuguaglycoside C (**7**) [7,21,27,28,29]. Structures of the isolated compounds and ^1^H- and ^13^C-NMR data are shown in Figure 2 and the Supporting Information, respectively. This is the first report describing the stereochemistry of the karaviloside VIII side chain to the best of our knowledge. Herein, we described the configuration of the stereocenters C-22 and C-23 through NOESY experiments as 22 *R* and 23*S* since H-22 showed correlation with H-20 and no correlation was observed between H-22 and H-23. Moreover, momordicosides A and L have been isolated from fruits of an Indian cultivar while the other compounds are mainly from Sri Lanka, China, and Japan cultivars. Therefore, this is the first finding of compounds **1**, **2**, **3** and **7** in fruits of the Indian cultivar. On the other hand, kuguaglycoside C was reported as an anticancer agent, showing significant cytotoxicity against human neuroblastoma IMR-32 cells [30]. In the same way, momordicoside A showed weak activity on glucose transport type-4 (GLUT4) on translocation cells [4] and momordicoside L was assayed as a hypoglycemic and antiproliferative compound, showing positive glucose uptake activity and no activity against human breast adenocarcinoma (MCF-7), human medulloblastoma (Doay), human colon adenocarcinoma (WiDr) and human laryngeal carcinoma (HEp-2).

### 2.2. Bioassays

#### 2.2.1. Anti-Inflammatory Activity

The anti-inflammatory activity of compounds **2**–**6** was analyzed. Compound **1** was assessed in our previous publication, and the low levels of compound **7** did not allow for further in-vitro assessments [18]. Interestingly, all the purified compounds exhibited anti-inflammatory properties by promoting the down-regulation of the pro-inflammatory gene markers *IL-6*, *TNF-α*, *COX-2,* and *iNOS* (Figure 3). All the compounds purified from the Indian bitter melon significantly decreased the expression of *IL-6* compared to LPS treated cells. The lowest expression of *IL-6* was observed in the cells treated with karaviloside VI, karaviloside VIII, momordicoside L, and momordicoside A. The downregulation of *iNOS* was also observed in all the compounds except momordicoside A. Similarly, momordicoside A, along with momordicoside L, significantly decreased *TNF-α* mRNA expression. Momordicoside L and karaviloside VI are also considerably reduced the expression of *COX-2* mRNA expression.

Pro-inflammatory cytokines mediate the inflammatory response in humans. LPS is a potent pro-inflammatory agent causing an increase in the expression of pro-inflammatory genes such as *IL-6*, *TNF-α*, *COX-2*, and *iNOS*. High levels of *IL-6* and *TNF-α* have been reported in patients with type-2 diabetes and insulin resistance [15]. Additionally, *COX- 2* has been reported to be induced under hyperglycemic conditions [16]. Any agent that promotes the decrease in the expression of these genes has the potential to work as a viable anti-inflammatory agent. Our results indicate that the compounds isolated from bitter melon may have potential antidiabetic activities by modulating the inflammatory process. In past studies, various bitter melon extracts and compounds have been reported to have anti-inflammatory activities in different cell models [21,31,32,33]. Bitter melon ethyl acetate extracts have been reported to decrease the expression of *iNOS*, *COX-2*, *IL-6,* and *TNF-α* in RAW 264.6 macrophages, but purified compounds derived from those extracts were not evaluated [34]. As such, the anti-inflammatory activity presented in this study suggests that bitter melon compounds are potential agents against inflammation and possible diseases that arise from chronic inflammation, such as diabetes.

#### 2.2.2. α-Amylase and α-Glucosidase

Several strategies have been explored in the management of diabetes mellitus for either reducing glucose production by the liver or enhancing insulin sensitivity or secretion [35]. Among these approaches, the inhibition of the *α*-amylase and *α*-glucosidase enzymes are directed to manage the post-prandial hyperglycemia by delaying starch hydrolysis by cleaving 1,4-glucosidic linkages [36,37]. In this sense, the α-amylase inhibitory effect of compounds **2**–**6** was also evaluated in this paper. The inhibitory effect ranges from 68.0 to 76.6%, but no significant statistical difference was observed among the compounds (Figure 4A). Additionally, the α-glucosidase inhibitory effect ranges from 23.7 to 56.5%. Karaviloside VIII was the most active compound, followed by karaviloside VI (40.3%), while the less active compounds were momordicoside L (23.7%), momordicoside A (33.5%) and charantoside XV (23.9%). Assayed compounds showed a lower α-glucosidase inhibitory effect than acarbose (Figure 4B).

### 2.3. Molecular Docking

Molecular docking studies were carried out to understand better the interactions of the isolated cucurbitane-triterpenes **2**–**7** with α-amylase and α-glucosidase as previously reported for a series of aglycone, monoglycoside cucurbitane-triterpenes, and 25ξ-isopropenylchole-5(6)-ene-3-*O*-β-d-glucopyranoside (**1**) isolated in this study [18].

#### 2.3.1. Molecular Docking Study with α-Amylase

Docking studies were carried out to understand the interaction of active compounds **2**–**6** inside the catalytic site of α-amylase. The crystalline structure of porcine pancreatic α-amylase (PDB ID: 1OSE) and active site amino residues are explained briefly in a previous publication [38]. The selection of the porcine pancreatic α-amylase crystal structure 1OSE in this study was based on three main reasons. First, we have used the porcine pancreatic α-amylase enzyme for our in vitro α-amylase inhibition study. Secondly, the selected protein forms a complex with the acarbose, which we used as a positive control or reference standard in our study. Finally, molecular models for the α-amylases from the human pancreas and human salivary are incredibly similar to the pig pancreatic molecular model. The homology modeling of the porcine and human pancreatic α-amylase is very similar (87.1%) compared with other amylases [39]. The binding energies of tested compounds with α-amylase ranged from −14.52 to −8.94 kcal/mol. The decreasing order of minimal binding energies in the case of α-amylase molecular docking studies are as follows: momordicoside A < kuguaglycoside C < karaviloside VIII < charantoside XV < karaviloside VI < momordicoside L (Appendix A). The docking results of the isolated compounds showed the binding site as the same as the binding sites for acarbose. Indeed, the docking analysis predicted that acarbose, a competitive inhibitor of α-amylase, was surrounded by Glu233, Asp300, and Asp197, which are the part of the catalytic residues of α-amylase. [40]. The molecular docking study for 25ξ-isopropenylchole-5(6)-ene-3-*O*-β-d-glucopyranoside (**1**) was previously reported in our study [18]. The binding modes in the active site of α-amylase of all purified compounds, except **1**, are shown in Figure 5A–F. The binding energy and number of hydrogen bonds of the five triterpenes against porcine pancreatic α-amylase are shown in Appendix A.

As shown in Figure 5A–F, all five triterpenes bound to the porcine pancreatic α-amylase through forming various hydrogen bonds. It was observed that the binding site for karaviloside VI (**2**) was close to the active site, allowing the interaction with Asp300, Ile235, and Trp59. Four hydrogen bonds were formed between compound **2** and the amino acid residues, including Glu240 (3 H-bonds), His201 (1 H-bond). The inhibition constant and binding energy for compound **2** is 18.79 nM and −10.54 kcal/mol (Appendix A).

In case of karaviloside VIII (**3**), the docking results predicted that the compound was enfolded in the catalytic domain adopting the same conformation as the acarbose site of α-amylase, as shown in Figure 5B. It was surrounded by fourteen key amino acid residues. The likeliest docked interactions of karaviloside VIII (**3**) and α-amylase is shown in Figure 5B. The ligand was surrounded by amino acid residues located in domain A, making hydrogen bonds with Lys200, His201, Trp59, and Asp356 (Appendix A). The principle interaction of karaviloside VIII (**3**) with α-amylase was surrounded by the key catalytic residue Trp59, and the interaction was crucial to inhibit the activity of α-amylase. Overall, the inhibition constant for compound **2** was 1.76 nM.

The refined docking of momordicoside L (**4**) showed weaker interactions than the other compounds because of the steric hindrance of the functional group at position 25, compared to compound **7**. The compound formed fewer H-bonds reflected in the lower binding energy of -8.94 kcal/mol. The 3D figure shows that momordicoside L interacted with sixteen key amino acid residues, including four conventional hydrogen bonds that were established between momordicoside L and Glu233, Ala307, His305 (Figure 5C). The inhibition constant was found to be 278.19 nM.

Figure 5D displays a 3D schematic interaction of momordicoside A (**5**) with α-amylase. Momordicoside A generated the best docking pose with a minimum binding energy of −14.52 kcal/mol, which indicates that momordicoside A showed the most stronger binding affinity with the α-amylase. Momordicoside A was found to anchor at the catalytic site of α-amylase by making ten conventional hydrogen bonds with Glu240, His101, Glu233, Asp197, Gly63 to residues resulting in potent inhibition of α-amylase. The strong inhibition activity of momordicoside A on α-amylase relies on the formation of multiple hydrogen bonds between hydroxyl groups and key residues of α-amylase. The molecular docking of momordicoside A provided supportive data for enzyme inhibition by predicting the binding site of α-amylase. We observed theoretical inhibition constant of 22.5 pM in the case of momordicoside A.

Analysis of molecular docking for the optimized conformation for charantoside XV (**6**) is shown in Figure 5E. The compound was bound to the active site of α-amylase with binding energy −10.64 kcal/mol. charantoside XV interacted with sixteen crucial amino acid residues in domain A of α-amylase Appendix A. The compound was able to interact with key amino acid residues, including Glu240, Gly306, by making three conventional hydrogen bonds. Besides, hydrophobic π-alkyl and alkyl interactions of charantoside XV with several amino acid residues, including Trr151, Lys200, Ala307, His201, Leu162, Val163, and Ile235. The inhibition constant of charantoside XV was 15.84 nM.

The refined docking of the kuguaglycoside C (**7**) with α-amylase is shown in Figure 5F. The binding energy for compound **7** in the active site of α-amylase is −11.54 kcal/mol. We observed that the compound **7** was able to interact with Asp242, Ser240, Leu246, Asn247, Ser282, Ala281, Asn302, Glu332, His280, Asp307, Thr310, Ser311, Lys156, Phe314, Leu313, Pro240 in the catalytic site of α-amylase (Figure 5F and Appendix A)**.** Also, the interactions calculated for the complex of kuguaglycoside C- α-amylase were effective because kuguaglycoside C (**7**) was buried entirely in the α-amylase binding pocket by forming ten hydrogen bonds with key amino acid residues, including Asp197, Glu233, His305, His299, and Asp300. The 2D figures for all compounds docked are compiled in supporting information (Appendix A).

Overall, the interactions between glucosides on the triterpenes and protein residues were suspected of playing an important role in determining the binding energy among triterpenes bearing the same backbone. Lower binding energy means that the ligand can more easily bind with the protein. Hence, regarding *α-*amylase molecular docking studies, we observed higher binding energy for momordicoside A and formed ten hydrogen bonds with the active site of *α-*amylase. Therefore, according to theoretical studies, momordicoside A was more likely to bind with α-amylase. However, in our in vitro studies, we did not observe a significant difference in the inhibitory activity, but we noticed momordicoside L showed comparatively higher inhibition among the six triterpenes. Therefore, our results suggested that in the inhibition process of triterpenes with α-amylase, lower binding energy does not necessarily lead to a higher inhibition activity, i.e., the inhibition activity of triterpene is affected by not only binding energy but also the chemical structure and glucoside type attached to a different position of triterpene. Besides, molecular docking studies are usually carried out under a theoretical vacuum condition, which deviates from real experimental conditions due to a low number of replications and prediction data [41].

#### 2.3.2. Molecular Docking Study with α-Glucosidase

The crystal structure of isomaltase from *Saccharomyces cerevisiae* (PDB ID: 3A4A) was used for the docking study. The 3D crystalline structures of α-glucosidase from *Saccharomyces cerevisiae* (maltase, EC 3.2.1.20) are unavailable in the PDB. However, there are crystalline X-ray structures of isomaltase or α-methylglucosidase have been deposited in the PDB. Previous studies have used isomaltase with a PDB ID 3AJ7 or 3A4A for molecular docking. This structure has a high-resolution X-ray structure, high sequence identity (72.51%) and sequence similarity score (0.54) with the *Saccharomyces cerevisiae* α-glucosidase MAL32 (UniProt entry P38158) [42,43]. Additionally, the authors identified several differences between residues lining the binding pockets of α-glucosidase and isomaltase, such as Phe157/Tyr158, Asp307/Glu204, Asp408/Glu411, Thr215/Val216, Ala278/Gln279, Val303/Thr306 or Ala178/Cys179, respectively. Most of these differences involve very similar residues. Hence we decided to use the crystal structure of isomaltase from *Saccharomyces cerevisiae* (PDB ID: 3A4A) for the molecular docking study with α-glucosidase.

The sequence alignment between α-glucosidase from bakers yeast (GI number 411229) and isomaltase (PDB ID: 3A4A) from *S. cerevisiae* have the structure identity and similarity of 73% and 85%, respectively. The key interactions of acarbose with *S. cerevisiae* α-glucosidase and *S. cerevisiae* isomaltase are quite similar.

The binding energy, interacting residues including H-bond interacting residues and Van der Waals interacting residues, along with the number of H-bonds with the crystal structure of isomaltase from *S. cerevisiae* for compounds **2–7,** are presented in Appendix A. Karaviloside VI (**2**) had a binding energy of −10.54 kcal/mol and occupied the active region of isomaltase by interacting with sixteen amino acid residues (Appendix A). In the conformation of isomaltase–karaviloside VI complex, the compound was able to establish four hydrogen bonds, including Glu332, Ala281, and Leu313. These hydrogen bonds overtly strengthened the interaction between karaviloside VI and isomaltase. The above interactions resulted in an inhibition constant of 12.48 nM. The 3D schematic interaction of karaviloside VI (**2**) is shown in Figure 6A. Indeed, the docking analysis of some cucurbitane triterpenes from Chinese bitter melon was carried out in our previous study [18,19] showed that compounds were surrounded by residues Glu277, His351, and Asp352, which are part of the catalytic residues of isomaltase.

The 3D schematic interaction of karaviloside VIII (**3**) is shown in Figure 6B. Karaviloside VIII was oriented toward the core of the binding pocket and interacted closely with the important seven amino acid residues in the active site (Appendix A). According to the Autodock 4.2 simulation results, the isomaltase–karaviloside VIII inhibitor complex showed −10.56 kcal/mol binding energy. Karaviloside VIII made six hydrogen bonds with Glu277 (two bonds), Gln279 (two bonds), Lys156, and Leu313. Moreover, two hydrogen bonds with Glu277 with a bond length of 1.4 and 1.5 Å, which is a key interaction to inhibit enzyme to a greater extent.

Momordicoside L (**4**) was surrounded by fourteen amino acid residues at the catalytic site of isomaltase. The 2D schematic and 3D interactions of momordicoside L and isomaltase are shown in Figure 6C. Momordicoside L generated the best docking pose with a minimum binding energy of −8.28 kcal/mol, and the inhibition constant of 852.19 nM. The ligand was surrounded by catalytic residue Asp307, and the interaction was likely crucial for inhibition of isomaltase. Eight conventional hydrogen bonds were observed between compound **4** and the isomaltase catalytic site residues, including Pro320, Pro312, His280, Ser304, Asn302, Ala281, and Glu232.

Similarly, momordicoside A (**5**) was bound to isomaltase’s active site with minimum binding energy −12.48 kcal/mol. Figure 6D clearly shows how momordicoside A interacted with twenty-one crucial amino acid residues (Appendix A). Compound **5** was able to establish six hydrogen bonds with Glu411, Ser314, and Glu332. The binding energy and inhibition constant for compound **5** are 706.7 pM and 12.48 kcal/mol, respectively.

The binding mode of charantoside XV (**6**) in the active site of isomaltase was shown in Figure 6E. The ligand was stabilized in an enzyme’s active site by interacting with nine amino acid residues (Appendix A). The compound was able to form seven hydrogen bonds with Pro312, Ser240 (two hydrogen bonds), Asp242 (three hydrogen bonds), and Asp307. Finally, the binding energy was −10.37 kcal/mol, and the inhibition constant was 25.02 nM.

The refined docking of kuguaglycoside C (**7**) with isomaltase generated the best pose with a minimum binding energy of −8.23 kcal/mol. The 3D schematic in Figure 6F shows that kuguaglycoside C established several hydrogen bonds within the isomaltase enzymatic pocket and interacted with sixteen amino acid residues (Appendix A). Three conventional hydrogen bonds were established between kuguaglycoside C and the active pocket of isomaltase, Ser282, Glu332, and Asp242. The 2D docking images of all compounds are shown in the Supporting Information Appendix A.

## 3. Materials and Methods

### 3.1. Chemicals

d-(+)-Glucose, d-allose, d-galactose, Ag_2_CO_3_, L-cysteine methyl ester hydrochloride, phenyl isothiocyanate, pyridine, *i*PrOH, MTBE, porcine pancreatic α-amylase, acarbose, dinitrosalicylic acid, ACN and MeOH HPLC grade, and *n*-hexane, CHCl_3_, MeOH, ACN, (CH_3_)_2_CO and EtOAc technical grade were purchased from Sigma-Aldrich (St. Louis, MO, USA). Nano-pure water HPLC grade (18.2 MΩcm) was obtained from a NANO pure purification system (Barnstead/Thermolyne, Dubuque, IA, USA). Silica gel 60 F254 TLC plates, HCl and H_3_PO_4_ were purchased from EMD Millipore, Inc. (Darmstadt, Germany). Silica gel 60 Å 40-63 µm and AcOH glacial were purchased from VWR International LLC (West Chester, PA, Switzerland), while RP-C_18_ Cosmosil 140 and formic acid were purchased from Nacalai Tesque, Inc., (Kyoto, Japan) and Alfa Aesar (Ward Hill, MA, USA), respectively. MeOH and pyridine NMR (perdeuterated) solvents were purchased from Cambridge Isotope Laboratories (Andover, MA, USA).

### 3.2. General Experimental Procedure

Optical rotations were measured in MeOH, using a Sac-i SACCHARIMETER instrument (ATAGO, Minato-ku, Tokyo, Japan. LC-HR-ESI-MS data was acquired using a maXis impact mass spectrometer (Bruker Daltonics, Billerica, MA, USA) coupled to a 1290 Agilent LC (Agilent, Santa Clara, CA, USA). The analysis of the fractions was performed using an Agilent HPLC 1200 Series (Agilent, Foster City, CA, USA) with a RP C_18_ Gemini column (250 × 4.6 mm; 5 µm) (Phenomenex, Torrence, CA, USA) using an adequate gradient with acetonitrile: water acidified. Chromatographic separations were performed on RP-C_18_ and silica cartridges and RP-C_18_ columns using a CombiFlash Rf flash chromatography (Combiflash Rf, Teledyne ISCO, Lincoln, NE, USA) equipped with a quaternary pump. Waters DeltaPrep preparative HPLC (Waters Delta Prep 4000; Waters, Milford, MA, USA) was used for final purification of certain compounds using a Phenomenex Gemini column (250 × 21.2 mm, 5 µm).

1D (^1^H; ^13^C and DEPT 135) and 2D NMR spectra [heteronuclear multiple-quantum correlation spectroscopy (gHMQC), double quantum filter correlation spectroscopy (DQ-COSY), Total correlation spectroscopy (TOCSY), heteronuclear multiple-bond correlation spectroscopy (gHMBC) and nuclear Overhauser effect spectroscopy (NOESY)] were recorded on a JEOL USA, Inc. spectrometer (Peabody, MA, USA) at 25 °C, operating at 400 MHz (^1^H) and 100 MHz (^13^C) using standard JEOL pulse programs. All samples were run in methanol-*d*_4_, except for **2** and **3,** which, because of insufficient solubility in methanol-*d*_4,_ were run in pyridine-*d*_5_. The chemical shifts are given in *δ* (ppm) and were referenced to residual solvent signals.

### 3.3. Plant Material

Fresh fruits of *Momordica charantia* (81 kg) were purchased from BCS Food Market, College Station, TX, USA. Bitter melons were cut into 0.5 inches size, seeds were removed manually, and the remaining pericarp was dried under shade, powdered using a home blender to get 60–80 mesh size, and stored at 25 °C until further use.

### 3.4. Soxhlet Extraction

Bitter melon powder (4.8 kg) was subjected to Soxhlet extraction using various solvents in a sequential manner with increasing polarities [18]. All the extracts were concentrated under vacuum separately and lyophilized to obtain dried powder as follows: *n*-hexane (57.6 g; 0.71%), ethyl acetate (91.8 g; 1.13%); acetone (161.9 g; 2.0%); methanol (131.5 g; 1.62%; methanol: water (1:1; *v*/*v*) (143.7 g; 1.77%) and methanol: water (1:3; *v*/*v*) (197.7 g; 2.44%). The extraction yields were expressed in the weight percentage of fresh fruit.

### 3.5. Purification Procedure

Flow chart of the isolation process of compounds **1**–**7** is presented in supporting information. Acetone extract (160 g) was impregnated with silica gel (~20 g) and loaded on an open column packed with 2 kg of silica gel. The extract was fractionated using step gradient elution using *n*-hexane, *n*-hexane: EtOAc (3:1, 1:1 and 1:3, *v*/*v*), EtOAc, EtOAc: MeOH (19:1, 9:1, 3:1, 1:1 and 1:3, *v*/*v*) to finish with MeOH. A total of 21 main fractions were pooled and dried under vacuum at 40 °C after analysis on silica gel TLC plates and sprayed with 10% H_2_SO_4_ in MeOH [18]. The solids from column fraction 1.11 (317.5 mg) were impregnated onto silica gel and fractionated by flash chromatography on a 40 g pre-packed cartridge and methyl *tert*-butyl ether (MTBE): acetone [(CH_3_)_2_CO] using flash chromatography. Fraction 2.4 (181 mg) was re-chromatographed using a 40 g packed RP-C_18_ column and ran using MeOH and MeOH: isopropanol (*i*PrOH) (9:1, *v*/*v*) to give fraction 3.3 (61 mg), which was precipitated to afford compound **1** (39 mg).

The resulting supernatant from fractions 1.11 (2.17 g) and 1.12 (11.9 g) was combined and impregnated onto silica and submitted to flash chromatography using a 120 g cartridge and a mobile gradient phase consisting of MTBE: (CH_3_)_2_CO. Fraction 4.10 (1.22 g) was submitted to similar chromatography conditions using EtOAc: (CH_3_)_2_CO. Fraction 5.5 (624 mg) was re-chromatographed on RP-C_18_ column (40 g) using MeOH: H_2_O: AcOH (92:8:0.1; *v*/*v*/%) to MeOH to furnish compound **2** (28 mg) from fraction 6.2.

The supernatant and solids from fractions 1.15 (8.7 g) and 1.16 (9 g) respectively were impregnated with silica gel and subjected to flash chromatography using a 330 g cartridge and gradient eluted with EtOAc: (CH_3_)_2_CO and (CH_3_)_2_CO: MeOH. Fraction 7.7 (8.8 g) was flash chromatographed using a RP-C_18_ column (40 g) with a gradient consisting of MeOH:H_2_O:AcOH (5:95:0.1 to 95:5:0.1 *v*/*v*/%). Fraction 8.7 (748 mg) was run in preparative HPLC using RP-C_18_ column and MeOH: H_2_O: formic acid gradient (5:95:0.1 to 95:5:0.1 *v*/*v*/%). Fraction 9.3 (55 mg) was re-chromatographed in RP-C_18_ preparative HPLC using ACN: H_2_O: formic acid (55:45: 0.1; *v*/*v*/%) to afford compound **3** (14 mg). Fractions 9.6 + 9.7 14–17 (136 mg) were subjected to RP-C_18_ preparative HPLC using ACN: H_2_O: formic acid (50:50: 0.1; *v*/*v*/%) to furnish compound **4** (13.9 mg) and compound **5** (31 mg). Fractions 8.5, 8.6, 7.4 and 7.6 were combined and ran using a 120 g RP-C_18_ cartridge and a MeOH: H_2_O: AcOH gradient. Fractions 11.8 (347 mg) and 11.10 (0.52 g) were individually re-chromatographed on RP-C_18_ preparative HPLC using ACN: H_2_O: formic acid (45:55:0.1; *v*/*v*/% and 60:40: 0.1) to give compound **6** (14.4 mg) and compound **7** (7.3 mg).

### 3.6. LC-HR-MS Analysis

LC-HR-ESI-MS were acquired using a maXis impact mass spectrometer (Bruker Daltonics, Billerica, MA, USA) coupled to a 1290 Agilent LC (Agilent) using a quadrupole time-of-flight mass detector equipped with an electrospray ionization interface controlled by Bruker software. Column fractions and purified compounds (2 µL) in MeOH were separated on Agilent Eclipse Plus RP-C_18_ (2.1 × 50 mm; 1.8µm) (Agilent) at 40 °C with a flow rate of 0.2 mL/min. The column was eluted with 0.1% formic acid in water (A) and 0.1% formic acid in acetonitrile (B) as follows, 2–95% B for 0–9 min, 95–2%B for 9–11 min, the column was re-equilibrated for 2 min before the next injection. All acquisitions were performed under positive ionization mode with a capillary voltage of 4200 V. Nitrogen was used as nebulizer gas (4.0 bar) and, as well as drying gas at 12 L/min, source temperature was maintained at 250 °C. Full scan mass spectra were acquired from *m*/*z* 50–2000. Data processing was done using Data Analysis Version 4.3.

### 3.7. Determination of the Absolute Configuration of the Sugar Unit

Compounds **6** (1 mg) was hydrolyzed with HCl (1 N) at 80 °C (1 mL) over 2 h, followed by a liquid-liquid partition with EtOAc (2 × 1 mL). The aqueous layer was neutralized with Ag_2_CO_3,_ and the supernatant was dried. L-cysteine methyl ester (2 mg) in pyridine (1 mL) was added and heated for 1 h at 60–70 °C. Phenylisothiocyanate (150 µL) was added and heated for an additional hour at 60–70 °C to form the thiocarbamoyl thiazoline derivative. The reaction mixture was analyzed by the HPLC method previously reported [44]. The absolute configuration of the sugar was determined by comparing the HPLC retention times of the prepared thiocarbamoyl thiazolidine derivatives to appropriate standards.

### 3.8. Physicochemical Parameters of Charantoside XV (**6**)

Amorphous white solid, [α]^25^_D_—71 (*c* 0.1, MeOH). HRESIMS of compound **1** showed a molecular ion at *m*/*z* 687.3967 [M + Na]^+^ (calculated 687.4047 *m*/*z* [M + Na]^+^ for C_37_H_60_O_10_). ^1^H- and ^13^C-NMR chemical shifts are presented in Table 1.

### 3.9. Bioassays

#### 3.9.1. Cell Culture and Quantitative Real-Time Polymerase Chain Reaction (qRT-PCR)

The anti-inflammatory activity of compounds purified from bitter melon was carried out using RAW 264.7 murine macrophage cells (ATCC, Rockville, MD, USA). The cells were cultured in RPMI 1640 medium supplemented with 10% (*v*/*v*) fetal bovine serum. Additionally, 100 U/mL of penicillin and 100 µg/mL streptomycin were added to the growth medium. Cultured cells were maintained at 37 °C in an incubator with 5% CO_2_. After 80% cell confluency, spent media was replaced with fresh media. For the analysis of anti-inflammatory gene expression, RAW 264.7 cells were seeded into 6 well plates (5.0 × 10^5^ cells/well). After a 24 h incubation period, the cells were treated with a 50 µM solution of purified compounds for 1 h followed by the addition of LPS (1 µg/mL) to all cells except the control cells. The concentration of compounds used in this assay was determined according to previously published research articles [45]. The cells were incubated for an additional 18 h after which the total RNA was extracted from the cells. Total RNA extraction was carried out using the Aurum Total RNA Mini Kit. The quantity of RNA extracted from the cells was determined using a Nanodrop Spectrophotometer. (Thermo-Fisher, Waltham, MA, USA) [19]. The purified RNA obtained from cell lysates were used as templates to synthesize cDNA. The synthesis procedure was carried out using the specified manufacture’s protocols (iScript cDNA Synthesis Kit, Bio-Rad Inc, Hercules, CA, USA).

Further, real-time PCR was carried out using the manufacturer’s specification for the Bio-Rad SYBR Green PCR Master Mix. The relative expression of *IL-6*, *TNF-α*, *COX-2*, and *iNOS* was compared and normalized to the expression of *GAPDH* from the respective treatment groups. Primer sequences for this study are available upon request. The negative control constituted of untreated cells, while the positive control consisted of LPS-stimulated cells.

#### 3.9.2. In Vitro α-Amylase Assay

The inhibition of α-amylase was measured using our previously published protocol using 96 well microplates [18]. Pure compounds (10 µL, 0.43 mM) and 140 µL of 1° saline were added to each well. Subsequently, 45 µL of 1% starch and α-amylase (10 mg/mL) was added. The reaction mixture was then incubated for 40 min at 25 °C followed by the addition of 50 µL of 3, 5-dinitrosalicylic acid (1%) in 20% Rochelle’s salt. The reaction mixture was incubated for 1 h at 50 °C, and the absorbance of each well plate was recorded at 540 nm. The absorbance of each reaction mixture was plotted in the analytical curve previously obtained for dextrose (25, 50, 75, 100, 125, 150, 200 µg/mL). The results were expressed as a percentage of inhibition.

#### 3.9.3. In Vitro α-Glucosidase Assay

The α-glucosidase inhibitory activity was evaluated according to a previously published protocol [19]. The assay mixture consisted of 70 μL of 100 mM phosphate buffer (pH 6.8), 10 µL (0.67 mM) of compound dissolved in DMSO, and 20 µL of 1 U/mL α-glucosidase solution were added to 96 well plates in triplicates. The plate was incubated at 37 °C for 15 min, followed by the addition of 20 μL of the p-nitrophenyl α-D-glucopyranoside substrate. The reaction mixture was incubated at 40 °C for 30 min, and then 50 μL of 0.1 M Na_2_CO_3_ solution was added. The absorbance was recorded at 405 nm using a microplate reader, and results were expressed in percentage of inhibition. All experiments were conducted in triplicate.

### 3.10. In Silico Docking Study

Purified compounds were docked with porcine pancreatic α-amylase (PDB ID: 1OSE) and α-glucosidase crystalline structure (PDB ID:3A4A) of isomaltase from *Saccharomyces cerevisiae*. The complex acarbose with porcine pancreatic α-amylase was removed using Biovia Discovery studio 4.5 software (Dassault Systems BIOVIA, Discovery Studio Modeling Environment, Release 4.5, San Diego, CA, USA, 2015). The methodology for the active site prediction, protein structure, and ligand preparations for molecular docking was provided in detail in our previous study [18]. Compounds were docked using the Lamarckian genetic algorithm in the AutoDock 4.2 Program (ADT, version: 1.5.6). The 2D visualization of ligand-protein interactions was analyzed using DS 4.5 (Dassault Systems), while PyMOL molecular graphics system (PyMOL Molecular Graphics System, San Carlos, CA, USA) was used to better visualize the 3D interactions between ligands and receptors.

### 3.11. Statistical Analysis

Data from α-amylase, α-glucosidase and anti-inflammatory assays were compared by ANOVA using GraphPad InStat software (San Diego, CA, USA). Differences among means were determined by the least significant difference Tukey’s post hoc test, with significance defined at *p* < 0.05 (*n* = 3 to 9).

## 4. Conclusions

The present study reports the purification and identification of a new 23-*O*-β-d-allopyranosyl-5β,19-epoxycucurbitane-6,24-diene triterpene (charantoside XV, **6**) along with five known cucurbitane-type triterpene glycosides and one sterol from crude acetone extract of an Indian cultivar of *Momordica charantia* L. var. *charantia*. Most of the isolated compounds downregulated the expression of pro-inflammatory *IL-6*, *TNF-α*, and *iNOS*, and mitochondrial marker *COX-2*. The most noteworthy activity was found in the downregulation of *IL-6* expression. With these findings, we evidence that the cucurbitane-type triterpene compounds could be responsible for the decreasing of the expression of defined markers. Further studies will incorporate western blot analysis to evaluate the induction of proteins related to inflammation in response to bitter melon triterpenes. Compounds also showed an antidiabetic potential by inhibition of the α-amylase and α-glucosidase carbolytic enzymes. Compounds docked as inhibitors of porcine pancreatic α-amylase complex showed binding energies from −14.53 to −8.94 kcal/mol and inhibition constants from 22.5 pM to 278.19 nM while when docked with α-glucosidase crystalline structure of isomaltase from *S. cerevisiae* binding energies ranged from −12.43 to −8.23 with inhibition constants from 706.7 pM to 929.04 nM.

## Figures and Tables

**Figure 1 molecules-26-01038-f001:**
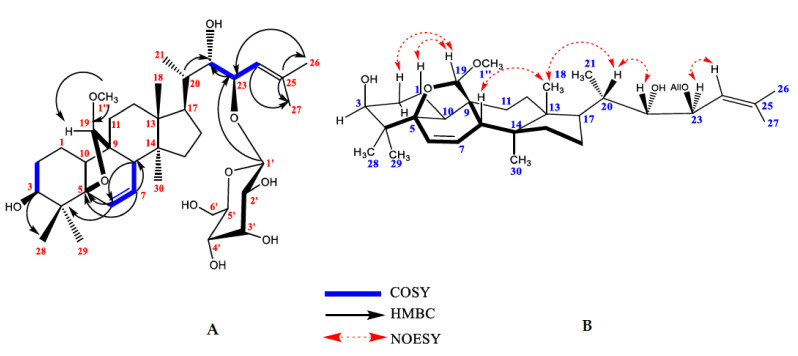
Crucial HMBC, ^1^H-^1^H COSY (**A**) and NOESY correlations (**B**) for compound **6.**

**Figure 2 molecules-26-01038-f002:**
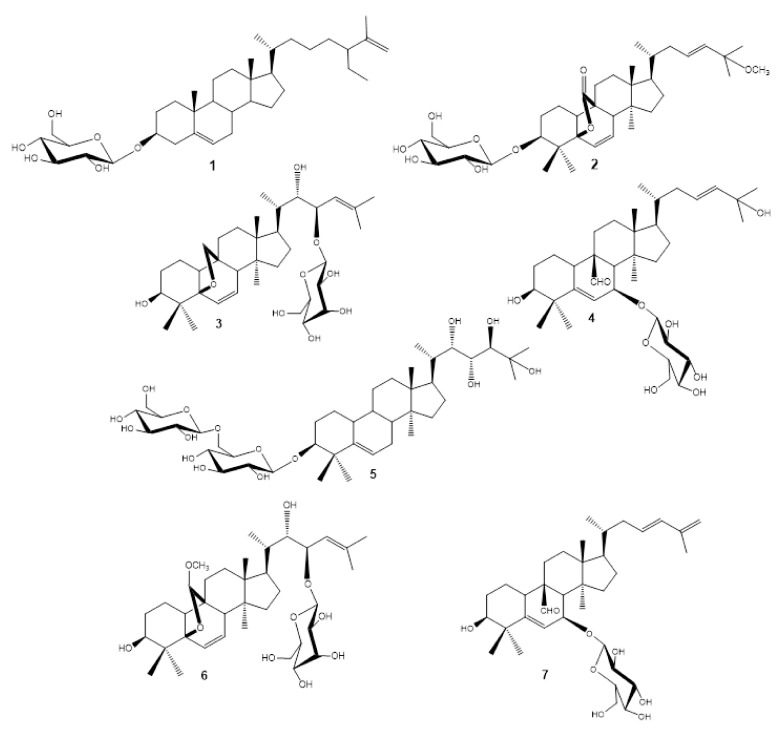
Chemical structures of compounds isolated and identified in the present study. (**1**) 25ξ-isopropenylchole-5(6)-ene-3-*O*-β-d-glucopyranoside, (**2**) karaviloside VI, (**3**) karaviloside VIII, (**4**) momordicoside L, (**5**) momordicoside A, (**6**) charantoside XV, and (**7**) kuguaglycoside C.

**Figure 3 molecules-26-01038-f003:**
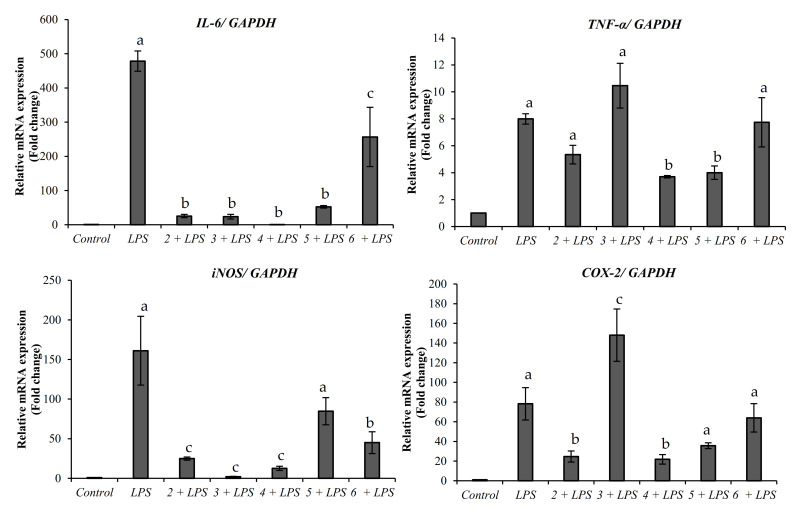
Effect of purified compounds on mRNA expression of *IL-6*, *TNF-α*, *iNOS*, and *COX-2* in LPS-induced murine macrophage RAW 264.7 cells. The cells were pretreated with 50 µM karaviloside VI (**2**), karaviloside VIII (**3**), momordicoside L (**4**), momordicoside A (**5**), and charantoside XV (**6**) followed by LPS (1 μg/mL) stimulation. Data are expressed as mean ± SD (*n* = 9) and analyzed by one-way ANOVA with a Tukey post hoc test. Different letters within the same plot indicate the significant differences at *p* < 0.05.

**Figure 4 molecules-26-01038-f004:**
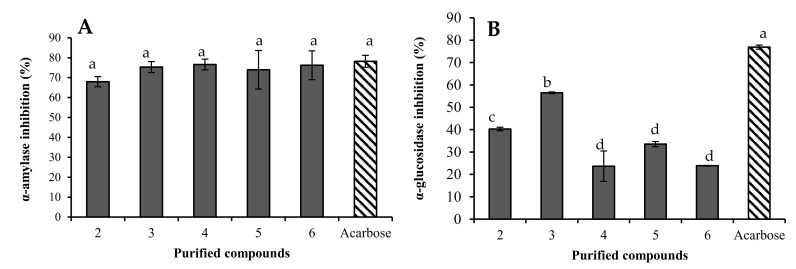
α-Amylase (**A**) and α-glucosidase (**B**) inhibitory effects of purified compounds: karaviloside VI (**2**), karaviloside VIII (**3**), momordicoside L (**4**), momordicoside A (**5**) and charantoside XV (**6**). Compounds were assayed at 0.67 mM and data are expressed as mean ± SD (*n* = 3), and analyzed by one-way ANOVA with a Tukey post hoc test. Different letters within the same plot indicate there were significant differences at *p* < 0.05.

**Figure 5 molecules-26-01038-f005:**
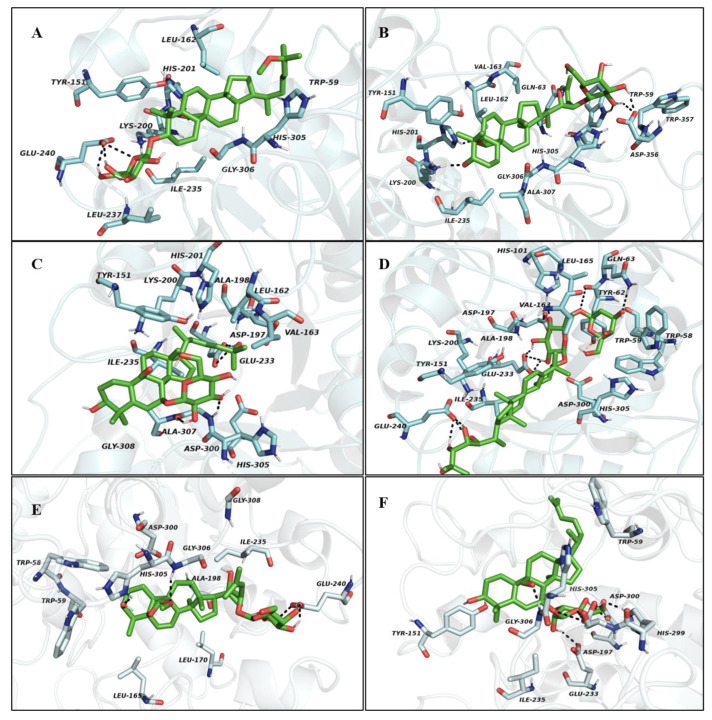
The 3D protein-ligand interactions for (**A**) karaviloside VI (**2**), (**B**) karaviloside VIII (**3**), (**C**) momordicoside L (**4**), (**D**) momordicoside A (**5**), (**E**) Charantoside XV (**6**), and (**F**) kuguaglycoside C (**7**) in the binding sites of α-amylase. Black dotted lines indicate hydrogen bonds between compounds and amino acid residues. Ligands in the active sites are denoted in green color. Active site residues are shown in cyans color.

**Figure 6 molecules-26-01038-f006:**
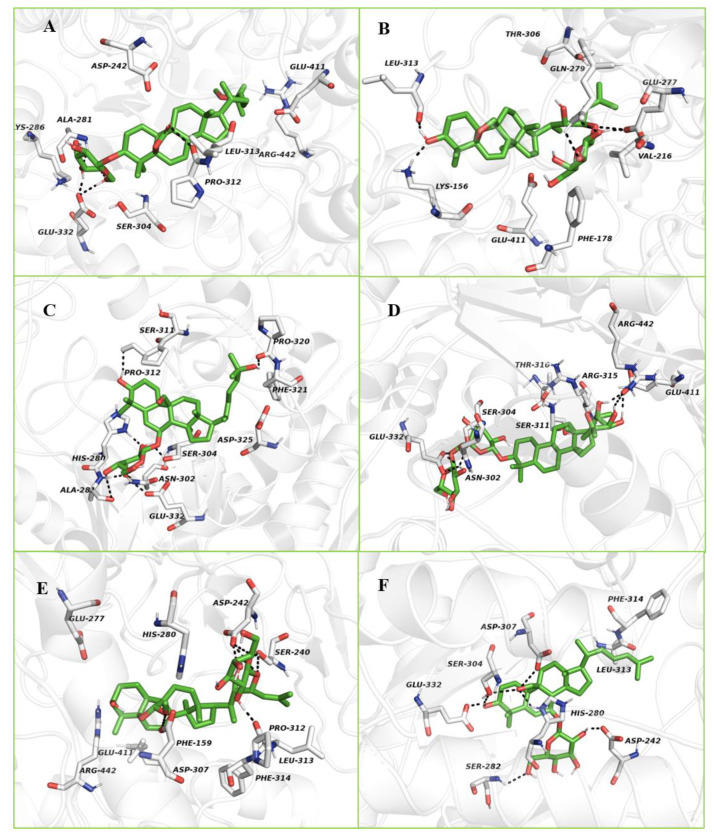
The 3D protein-ligand interactions for (**A**) karaviloside VI (**2**), (**B**) karaviloside VIII (**3**), (**C**) momordicoside L (**4**), (**D**) momordicoside A (**5**), (**E**) charantoside XV (**6**), and (**F**) kuguaglycoside C (**7**) in the binding sites of isomaltase. Black dotted lines indicate hydrogen bonds between compounds and amino acid residues. Ligands in the active sites are denoted in green color. Active site residues are shown in white color.

**Table 1 molecules-26-01038-t001:** ^1^H and ^13^C NMR chemical shifts of compound **6**.

Position	Compound 6 (MeOH-*d*_4_)	
	*δ* _H_	*δ* _C_
1	1.15; 1.50 m, 2H	18.5
2	0.82; 1.84 m, 2H	28.3
3	3.36 m, 1H	77.8
4	−	38.5
5	−	88.1
6	5.98 dd (*J* = 9.8, 2.1 Hz, 1H)	132.2
7	5.58 dd (*J* = 9.8, 3.7 Hz, 1H)	133.8
8	2.86 m, 1H	43.2
9	−	49.7
10	2.44 m, 1H	42.1
11	1.54; 1.75 m, 2H	24.3
12	1.54; 1.54 m, 2H	32.0
13	−	46.5
14	−	49.0
15	1.32; 1.32 m, 2H	35.0
16	1.34; 1.34 m, 2H	29.0
17	1.97 m, 1H	47.7
18	0.84 s, 3H	14.8
19	4.70 s, 1H	113.3
20	1.77 dd (*J* = 11.3, 1.0 Hz, 1H)	42.1
21	0.96 d (*J* = 6 Hz, 3H)	14.7
22	3.63 m, 1H	77.3
23	4.29 m, 1H	81.5
24	5.24 d (*J* = 10.2 Hz, 1H)	124.6
25	−	137.4
26	1.74, 3H	26.7
27	1.77, 3H	19.1
28	1.18 s (3H)	20.9
29	0.87 s (3H)	24.6
30	0.87 s (3H)	20.6
23-*O*-All		
1′	4.73 d (*J* = 8 Hz, 1H)	102.6
2′	3.32, 1H	73.1
3′	4.05, 1H	73.0
4′	3.48, 1H	68.8
5′	3.58, 1H	75.7
6′	3.60; 3.76, 2H	63.1
1″	3.41, 3H	58.2

## Data Availability

The data presented in this study are available in Appendix A.

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
