# Peer review of "Anti-Inflammatory, Antidiabetic Properties and In Silico Modeling of Cucurbitane-Type Triterpene Glycosides from Fruits of an Indian Cultivar of Momordica charantia L."

_molecules, 2021, doi:10.3390/molecules26041038_

Round 1
Reviewer 1 Report
This manuscript includes lots of work to identify the natural products from Momordica Charantia. However, the procedures and results to elucidate the binding condition between α-Amylase and α-Glucosidase, and isolated compounds are not described clearly.
Docking study with α-amylase followed the previous publication, references 38 where 1ose.pdb was selected for the 3D structure of α-amylase. More than a thousand structures are deposited in the protein data bank. references 38 did not mention the reason why 1ose.pdb was not selected among many 3D structures of α-amylase in the PDB. Likewise, this manuscript did not explain the reason why the 3D structure of α-amylase was chosen.
In the case of α-glucosidase, the authors used the 3D structure obtained based on the homology modeling with isomaltase (PDB ID: 3A4A). Many 3D structures of α-Glucosidase are deposited in the PDB. Why didn’t the authors select one of them?
Before publication, two basic problems mentioned above should be clarified. If the reason for the selection of the 3D structures of the target proteins is not reasonable, whole docking results cannot be accepted. I like to suggest the 3D structures of the target proteins should be selected based on the clear reason and in silico docking part should be written again.
Reviewer 2 Report
Its a nicely written manuscript. I have a few minor comments/concerns:
- Please correct language. The grammar is OK but senetences like : "Our results indicate that the compounds isolated from bitter melon may be excreting their potential antidiabetic activities by hindering the inflamma-tory process." Need to be rephrased.
- I would suggest doing a Western Blot in future and also mentioning this as furture work.
- My last concern is: Since the bitter melons were bought in the "BCS Food Market, College Station, TX", how sure are the authors about the species and genus of the sample? Did they do any typing or botanical classification? How did they enure that all the melons in 81kg are of same variety.
Reviewer 3 Report
The authors describes the identification, purification and study of the anti-inflammatory, antidiabetic properties of of cucurbitane-type triterpene glycosides from fruits of Indian cultivar of Momordica charantia L, as well as the in silico modeling of their interaction with active sites in enzymes to provide insights on their bio activities.
The topic is of interest to the journal audience.
The research is very well-conceived and executed.
Conclusions are well-supported by the data.
English is generally fine.
I didn't detect any critical scientific flaws.
I just invite the authors to:
- check that temperatures and yields are reported as follows "X °C" and "Y%", respectively
- be more precise with yield and tells whether they're w% or mol%
- verify that the number of H is given in every lines in Table 1
- move the following sentence " Flow chart of the isolation process of compounds 1-7 is presented in supporting information. " at the beginning of the 3.5 section for better clarity
- provide melting point every time the compound is a solid
In conclusion, I recommend the publication of this manuscript after minor revisions.
Round 2
Reviewer 1 Report
This manuscript is acceptable as it is.